# Peptidoglycan Endopeptidase from Novel Adaiavirus Bacteriophage Lyses *Pseudomonas aeruginosa* Strains as Well as *Arthrobacter globiformis* and *A. pascens* Bacteria

**DOI:** 10.3390/microorganisms11081888

**Published:** 2023-07-26

**Authors:** Karel Petrzik

**Affiliations:** Institute of Plant Molecular Biology, Biology Centre of the Czech Academy of Sciences, Branisovska 1160/31, 370 05 Ceske Budejovice, Czech Republic; petrzik@umbr.cas.cz

**Keywords:** adaiavirus, endolysin, host range, G+ and G− activity

## Abstract

A novel virus lytic for *Pseudomonas aeruginosa* has been purified. Its viral particles have a siphoviral morphology with a head 60 nm in diameter and a noncontractile tail 184 nm long. The dsDNA genome consists of 16,449 bp, has cohesive 3′ termini, and encodes 28 putative proteins in a single strain. The peptidoglycan endopeptidase encoded by ORF 16 was found to be the lytic enzyme of this virus. The recombinant, purified enzyme was active up to 55 °C in the pH range 6–9 against all tested isolates of *P. aeruginosa*, but, surprisingly, also against the distant Gram-positive micrococci *Arthrobacter globiformis* and *A. pascens*. Both this virus and its endolysin are further candidates for possible treatment against *P. aeruginosa* and probably also other bacteria.

## 1. Introduction

*Pseudomonas aeruginosa* is a Gram-negative polytrophic bacterium of great importance for human and animal health. It is considered an opportunistic nosocomial pathogen commonly associated with pneumonia, skin, soft tissue, eye, urinary tract, and otitis diseases [1,2], and it is extremely dangerous to immunocompromised and immunodeficient patients and patients with cystic fibrosis [3]. Recently, many *P. aeruginosa* isolates have shown resistance to almost all classes of antibiotics, making conventional therapies with combined antibiotics increasingly ineffective [4]. New antibiotics effective against a broad spectrum of multidrug-resistant *P. aeruginosa*, such as ceftolozane–tazobactam, ceftazidime–avibactam, doripenem, and plazomicin, have become available in recent years, but they might have some unnecessary adverse effects or be restricted solely to intravenous use [5]. In addition, the development of new antibiotics is costly and time-consuming and time and again has failed to halt the accelerated development of multidrug-resistant bacterial strains. Going after the bacteria with appropriate bacteriophages, which are natural bacterial enemies, or using antimicrobial enzymes could be new therapeutic strategies to combat *P. aeruginosa* infections [6,7,8].

GenBank lists about 170 viruses (bacteriophages) with genome sizes between 3 kbp and 309 kbp using *P. aeruginosa* as a propagation host (accessed June 2023). Thus far, as is known, many of the lytic viruses have limited host ranges and preferentially replicate in one or a few related strains [9,10]. The use of a phage cocktail could compensate for this limitation, and a clinical trial treating burns infected with *P. aeruginosa* was conducted in this way without adverse effects [11]. In recent years, recombinant bacteriophage endolysins (lysins) have been produced and tested as promising alternative antibacterial agents to combat *P. aeruginosa* in medicine [12,13]. These address some concerns and limitations related to uncontrollable dosing, immunogenicity, the potential for horizontal transmission, emerging resistance, regulatory hurdles, and, in some countries, use of intellectual property on complete bacteriophages [14,15]. Endolysins are enzymes used by bacteriophages to enzymatically degrade the peptidoglycan layer of the bacterial host, thereby leading to cell lysis and the release of virion progeny [16]. The endolysins have been identified in both lytic and lysogenic bacteriophages and, like PlyE146, gp144, and LysPA26 [17,18,19], they are effective against a broader range of bacterial strains and species, have potential to destroy biofilms, and could be used in synergy with antibiotics.

The endolysin of the novel Pseudomonas virus Hadban is most likely a peptidoglycan endopeptidase. It has its counterpart in Arthrobacter phage Adaia, with 96.4% amino acid (aa) sequence identity, but no other similar genes are present in GenBank (April, 2023). In the presented work, the optimal reaction condition of this enzyme was evaluated.

## 2. Materials and Methods

A new *P. aeruginosa*-specific phage was isolated from soil when it was enriched overnight with a culture of *P. aeruginosa*-isolate POCH2. Plaques that developed on a soft agar plate with the POCH2 isolate were purified in three passages, propagated, then concentrated from the bacterial lysate with 10% PEG 6000 and 4% NaCl precipitation [20]. The host specificity of the new phage was determined by spotting 1 µL of the purified phage onto a soft agar layer containing the putative host, culturing at 27 °C for 16 h, and then evaluating. For a one-step growth curve, a bacterial culture was grown to OD_600_ = 0.3 (approximately 10^8^ CFU/mL) and inoculated with the phage to obtain a multiplicity of infection > 1. Adsorption was performed for 10 min at 27 °C, followed by incubation on a shaker at the same temperature. Samples were taken every 20 min, diluted, plated onto lawns of the susceptible *P. aeruginosa* POCH strain using the double agar overlay assay, and then evaluated after 16 h.

DNA for sequencing was extracted from purified viruses after RNase, DNase, and proteinase K treatment as described previously [20], then sequenced as paired-end reads on the Illumina platform by Eurofins (Konstanz, Germany). The separate reads thus obtained were demultiplexed, adapter trimmed, and then de novo assembled using CLC Genomics Workbench 8.5.1. software (Qiagen, Hilden, Germany). The virus-like sequence was identified with BLAST. The coding regions were predicted using GeneMark v3.42 [21] and annotated using RAST [22]. HHpred was used to assess protein function [23]. Nucleotide and protein sequence comparisons and phylogenetic analyses were performed in MEGA X [24] and VipTree 3.5 [25].

The endolysin gene was amplified with the primers 5′-cacc**ATG**ACGAAGTTCAGTAGCC and 5′TTAGACCTTTACGCCCATG (the initial codon is in bold and termination codon is underlined) with Sapphire 2× Premix (Takara Inc., Kusatsu, Japan), cloned into the pET100 TOPO plasmid (Fisher Scientific, Carlsbad, CA, USA), then transformed into *E. coli* TOP10 cells. Clones with verified plasmid and inserts were propagated. The plasmid was then extracted and used for transformation of *E. coli* BL21(DE3) cells. Protein expression was induced for 16 h at 30 °C with 0.5 mM IPTG according to the manufacturer’s instructions. The protein was extracted from the bacterial lysate using a His-Trap column (GE Healthcare Ltd., Little Chalfont, UK), dialyzed against 50 mM phosphate buffer, then freeze-dried.

Endolysin activity was detected using a turbidity reduction assay in which 20 µg of the purified endolysin in 50 µL volume was added to 150 µL volume of *P. aeruginosa* cells after chloroform permeabilization, as described by Briers et al. [26], and then measured in intervals at A_620_ nm. The experiment was performed three times independently. To determine the optimal temperature, the purified enzyme was incubated in 20 mmol/L phosphate buffer at 30, 40, 50, 55, 60, and 70 °C with permeabilized *P. aeruginosa* POCH2 cells for 60 min. To determine the optimal pH for endolysin activity, the enzyme was incubated with *P. aeruginosa* POCH2 cells in 20 mmol/L phosphate buffer at pH values between 4 and 10. Relative activity was determined by comparing the maximum activity of the dataset and the lytic activity of each assay [27].

## 3. Results and Discussion

The POCH2 isolate of *P. aeruginosa* was obtained from the ear swab of a dog with otitis media and identified using 16S rDNA sequencing. This strain was resistant to β-lactam antibiotics, valinomycin, and novobiocin, but was sensitive to chloramphenicol, kanamycin, tetracycline, and neomycin. The bacteriophage lysing this strain was purified and the purified virions had an isometric head with a diameter of 61 nm (±2.4 nm) and a non-contractile tail with a length of 184.4 ± 8.7 nm (*n* = 31), reflecting siphoviral morphology (Figure 1). The latent period of the bacteriophage on POCH2 was approximately 3 h and the average burst size was calculated to be 15 PFU/infected cell. We provisionally named this bacteriophage Pseudomonas virus Hadban.

A single viral contig with a length of 16,449 bp was assembled using CLC software from about 400,000 reads with an average coverage of 3672×. The genome of Hadban was 16,449 bp long, the G+C content was 56.2%, and 28 genes predicted by RAST were arranged on one strand. No tRNA genes were identified in Aragorn. In addition, the genome of Hadban had 10 nt 3′-cohesive termini (CCCGCGCCCC) identical to those of the Arthrobacter virus Adaia and similar to those of some decurroviruses (PhagesDB). Eleven genes at the 5′ end encoded structural proteins, the product of ORF 16 was predicted to be viral endolysin, and ORF 28 contained an HNH endonuclease domain (pfam14279) similar to the endonucleases of gordoniaviruses, adaiavirus, and atraxavirus (Figure 2, Appendix A). No genes associated with DNA replication or metabolism were identified on the genome.

The Arthrobacter phage Adaia with genome size 15,840 bp and G+C content 56.1%, the Arthrobacter phage Atraxa with genome size 14,927 and G+C content 58%, and decurroviruses with genome sizes ranging from 14,830 to 15,556 bp and G+C contents around 60.1% were found to be the viruses most similar to this new virus [28]. Phylogenetic analysis of the complete genomes assigned our virus to the genus Adaiavirus, with 90.4% identity to the type virus. Recombination analyses of the aligned complete sequences of virus Hadban, Adaia virus, Atraxa virus, and selected decurroviruses performed with the RDP5 program [29] failed to detect recombination of Hadban using any algorithm. Therefore, we concluded that Pseudomonas virus Hadban is a new virus of the genus Adaiavirus (Figure 3).

The host range of the virus was investigated on verified *P. aeruginosa* strains from the Czech Collection of Microorganisms (CCM, Brno, Czech Republic), from the Collection of Phytopathogenic and Agriculturally Beneficial Bacteria (CPABB, Praha-Ruzyně, Czech Republic), and from the German Collection of Microorganisms and Cell Cultures (DSMZ, Heidelberg, Germany). The complete bacteriophage lysed all *P. aeruginosa* strains, but not the atypical brown color-producing strain CCM 3630 and strain DSM 22644. The extraordinarily high amino acid sequence identity of the Pseudomonas virus Hadban and Arthrobacter virus Adaia in several genes, including endolysins (96.4%), prompted us to also test the lytic properties of the new bacteriophage on two *Arthrobacter* species. Surprisingly, the bacteriophage lysed both *Arthrobacter pascens* CCM 1653 and *Arthrobacter globiformis* CCM 193 strains (Table 1), but it did not lyse strains of *Dickeya* sp., *Stenotrophomonas maltophilia*, or *Staphylococcus pseudintermedius*. To date, there is no information on the host specificity of decurroviruses, adaiavirus, or atraxavirus. It should be emphasized that the hosts of all Hadban-related bacteriophages are Gram-positive *Arthrobacter* sp. bacteria (Micrococcaceae, Actinomycetales, Actinobacteria) and the host of virus Hadban is a Gram-negative *P. aeruginosa* (Pseudomonadaceae, Gammaproteobacteria). The proteins involved in virus Hadban’s ability to infect *P. aeruginosa* bacteria are not known. We can speculate that the tail protein(s) responsible for binding the virus to the bacterial cell wall have been altered. From this point of view, the minor tail protein (product of ORF 14) is a good candidate, because this protein differs greatly in its C-terminal part and has a 74% overall aa identity with the corresponding protein of virus Adaia, while the other structural proteins are 95% to 100% identical to the corresponding proteins (see Appendix A). In addition, it is possible that yet-unidentified hypothetical proteins encoded by ORFs 15, 20, and 26 play a role. Although the Hadban virus was purified from the plaques by three passage cycles, we cannot exclude contamination at this point with some Gram-positive-specific phage/s that could contribute to the lysis of the intact *Arthrobacter* sp. cells.

Endolysins from Gram-positive bacterial hosts usually have a cell-wall-binding domain and an enzymatic active domain, whereas those from Gram-negative hosts usually have no specific cell-wall-binding module [30]. There are some exceptions, however, such as the lysins KZ144 and EL188 from *Pseudomonas* phages, which also have a modular structure with an N-terminal cell-wall-binding domain and a C-terminal enzymatically active domain [16].

The ORF16 of virus Hadban encodes a protein 221 aas long (23.3 kDa) in whose N-part within amino acid positions 4–142 a domain of the M23 family metallopeptidase was found and in whose C-part within amino acid positions 180–214 a putative peptidoglycan-binding protein (pfam01471) was found. This protein is, therefore, presumably an endolysin of this virus. The putative endolysin of the related Arthrobacter virus Atraxa (ORF 15) has a similar domain structure, but it is smaller (a protein 207 aas long) and has an amino acid sequence identity of 47.3% with the endolysin of virus Hadban. Decurroviruses’ endolysins are 149 aas long and have the metallopeptidase domain only [31]. The recombinant endolysin of virus Hadban (Figure 4a) was active with all *P. aeruginosa* strains tested, including the strains CCM 3630 and DCMZ 22644, which were not lysed by the complete virus (Figure 4b). There were no significant differences between the strains. It also lysed both *Arthrobacter* strains under the same experimental conditions. The enzyme was active up to 55 °C, although its activity at 55 °C was only about 60% of the maximum activity reached at 50 °C (Figure 4c), and it functioned in the pH range of 6–9 (Figure 4d).

## 4. Conclusions

Endolysins generally have broader specificity than do whole phages [19,32,33]. They act rapidly, are stable over wide pH and temperature ranges [15], and can be used against biofilms [19] and in combination with antibiotics or other antibacterial enzymes [16]. Here, we have described not only the novel bacteriophage that lyses *P. aeruginosa* per se, but also its unique ability to lyse completely different, unrelated bacteria.

## Figures and Tables

**Figure 1 microorganisms-11-01888-f001:**
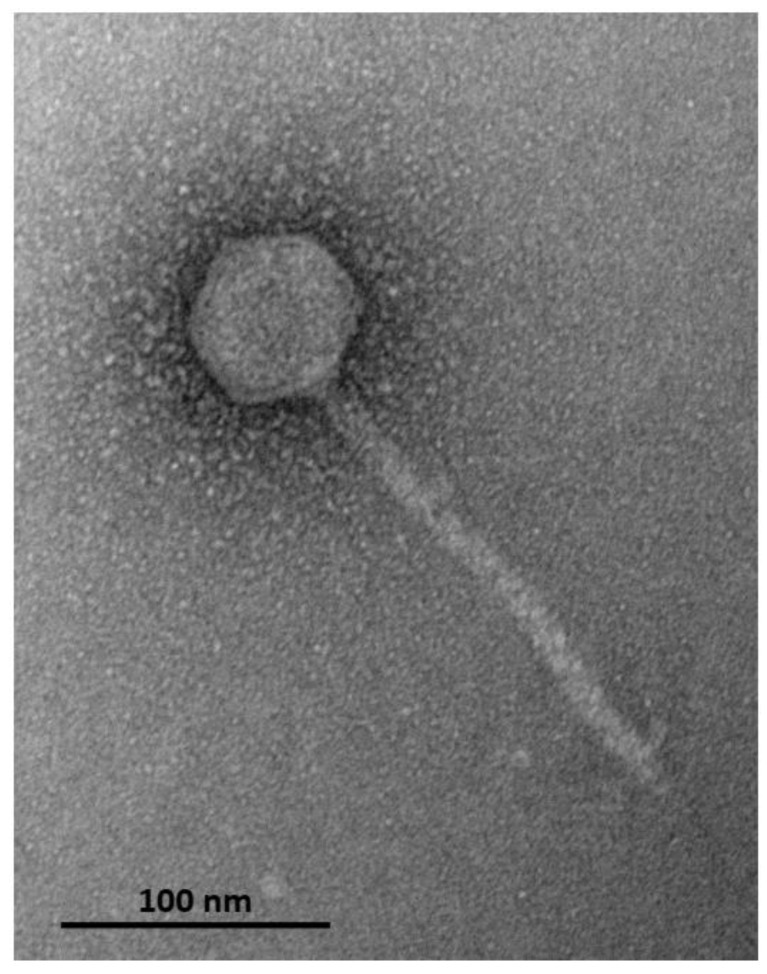
Electron microscopy of purified Pseudomonas virus Hadban negative-stained with 2% (*w*/*v*) uranyl acetate and examined with a JEM-1400 JEOL transmission electron microscope.

**Figure 2 microorganisms-11-01888-f002:**
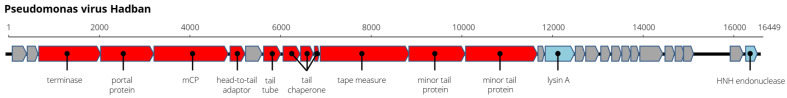
Genome arrangement of the Pseudomonas virus Hadban. Structural genes are marked red, genes with unknown function are marked gray.

**Figure 3 microorganisms-11-01888-f003:**
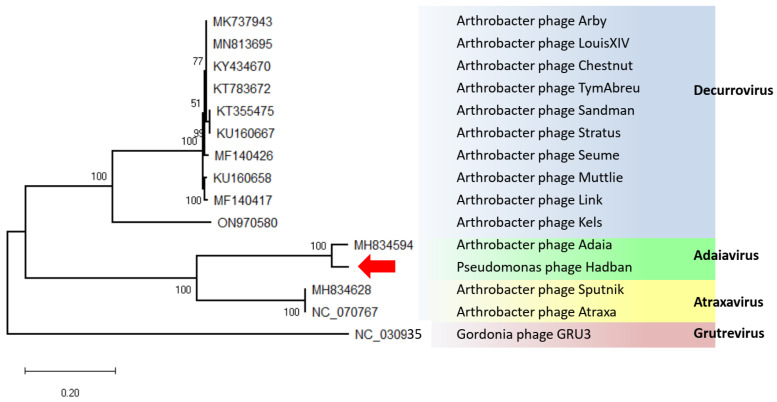
Maximum-likelihood tree computed from complete genome sequence alignment of virus Hadban (marked with arrow), adaiaviruses, atraxaviruses, and selected gordoniaviruses.

**Figure 4 microorganisms-11-01888-f004:**
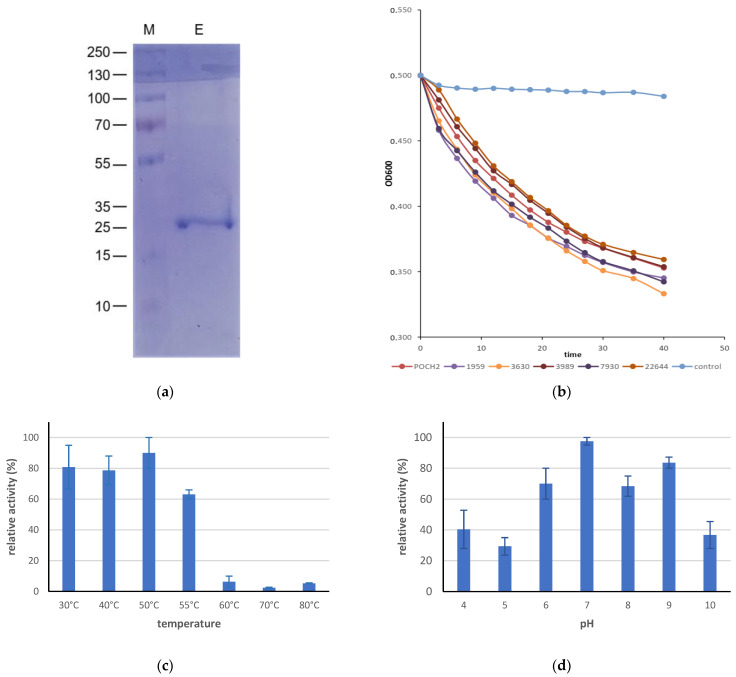
(**a**) SDS-polyacrylamide gel separation of purified virus Hadban endolysin. M-protein size is pre-stained marker (Fisher Scientific), endolysin. (**b**) Turbidity reduction assay of the recombinant virus Hadban endolysin with distinct *P. aeruginosa* strains. (**c**) Optimal temperature for the bactericidal activity of the endolysin after 1 h incubation at given temperature. (**d**) Optimal pH for the bactericidal activity of endolysin. The assays were performed in triplicate, and error bars represent standard deviations.

**Table 1 microorganisms-11-01888-t001:** Host specificity of the Pseudomonas virus Hadban.

Strain	Virus	Endolysin
*P. aeruginosa* POCH2	+++	+++
*P. aeruginosa* CCM 1959	+++	+++
*P. aeruginosa* CCM 3630	−	+++
*P. aeruginosa* CCM 3989	+	+++
*P. aeruginosa* CCM 7930	+	+++
*P. aeruginosa* DSMZ 22644	−	+++
*Arthrobacter pascens* CCM 1653	+++	+++
*Arthrobacter globiformis* CCM 193	+++	+++
*Dickeya* sp. CPABB 050	−	−
*Stenotrophomonas maltophilia* CCM 1640	−	−
*Staphylococcus pseudintermedius* DSMZ 25714	−	−

One µL of purified phage was spotted on soft agar with corresponding bacterium. Clear lytic zone observed after 16 h is marked +++, opaque zone is marked +, and no lytic zone is marked −.

## Data Availability

The complete genome sequence of Pseudomonas virus Hadban has been deposited in GenBank under accession number OR067381.

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
