# Peer review of "Peptidoglycan Endopeptidase from Novel Adaiavirus Bacteriophage Lyses Pseudomonas aeruginosa Strains as Well as Arthrobacter globiformis and A. pascens Bacteria"

_microorganisms, 2023, doi:10.3390/microorganisms11081888_

Round 1

Reviewer 1 Report

Opinion on Karel Petrzik: ” Peptidoglycan endopeptidase from novel Adaiavirus bacteriophage lyzes Pseudomonas aeruginosa strains as well as Arthrobacter globiformis and A. pascens bacteria”

This manuscript describes the isolation, sequencing and characterization of a novel phage of P. auruginosa. The phage (referred to as Pseudomonas virus Hadban) morphologically belongs to siphoviridae, and is a close relative of Adaia virus, Atraxa virus and decurroviruses. The endolysin gene of the novel phage was identified, cloned and expressed in E. coli, purified, and its lytic activity was tested on multiple species. Surprisingly, it lyses even those Pseudomonas strains that are resistant to the novel phage, as well as the distant Gram-positive micrococci Arthrobacter globiformis and A. pascens. The domain structure of the new lysin resembles more that of the lysins of Gram+ bacteria.

Questions:

1. Why were the phage target assays and phage growth experiments carried out at 27 oC?

2. The authors state that the novel phage did not lyse strains of Dickeya sp., Stenotrophomonas maltophilia, or Staphylococcus pseudintermedius. What about the lysin, was it active on these strains?

Comments:

Concerning that a pET vector was used to express the lysin, can we assume that the E. coli host was not BL21, but BL21(DE3)? Please correct, if so.

The description of the transmission electron microscopy experiment is missing from the Methods section.

Certain abbreviations are not defined, e.g. RDP5, ML, mCP.

Line 180: “which did not lyse the complete virus” -> which were not lysed by the complete virus”

“lyzes” -> “lyses”

lysines -> lysins

Reviewer 2 Report

The manuscript Peptidoglycan endopeptidase from novel Adaiavirus bacteriophage lyzes Pseudomonas aeruginosa strains as well as Arthrobacter globiformis and A. pascens bacteria written by Karel Petrzik deals with a brief description of the new Hadban phage as well as its endolysin. The study includes not only TEM images, but also the whole genome sequence of the phage published in the Genbank database. In addition, the author identified, cloned and studied endolysin in the work. Despite the well-described basic characterization of phage and endolysin, the work has several flaws. Unfortunately, the author chose a short article (communication) and was thus unable to describe several necessary facts. I have several comments about the manuscript.

According to the data presented, Hadban phage should be able to infect and lyse both G+ and G- bacteria (Table 1). It is not very likely that a small phage with a genome of only 16.5 kb and without its own DNA metabolism genes would be able to infect such different groups of bacteria. I consider phage purification methods insufficient. More evidence would need to be presented to confirm that it is not a mixture of phages. For example, it is not known what other contigs the author obtained during de novo assembly. Alternatively, he used a single extracted plaque without amplification to determine the range of hosts.

Did the author obtain a plaque when determining the host specificity? What was their morphology?

In the methods, the author describes the measurement of phage adsorption, but does not indicate what the adsorption was on the strain that the phage does not infect (CCM3630, DSMZ 22644).

When measuring endolysin activity, you used cells treated with chloroform. Was the control also processed this way? In the case of G+ bacteria, endolysin should work even without processing the cells. Therefore, it would be appropriate to measure its activity in such a case.

In the discussion, it would be appropriate to comment on the type of peptidoglycan and better explain how it is possible that the enzyme infects both G+ and G- bacteria

In the discussion, the author did not comment on the decrease in endolysin activity at pH8 (Fig 4d)

The manuscript has a significant number of shortcomings, but if the author adds proof of the purity of the phage lysate and endolysin activity on untreated G+, I consider the work publishable.

Round 2

Reviewer 2 Report

Dear Author,

Thank you for the replies to my review. I agree with most of them. I also like the plan to study phage and endolysin further. Nevertheless, I believe it would be appropriate to discuss the possibility that one of the phages capable of infecting G+ bacteria remained in the Hadban phage lysate. In your answer, you mentioned phages infecting G+ bacteria such as Staphylococcus and Cytobacillus. Moreover, based on personal experience, I know that phage passaging is often insufficient. In my case, ultracentrifugation in CsCl was necessary.

Author Response

Dear reviewer, we have added a sentence expressing the existing doubts about the purity of the Hadban virus preparation.

"Although Hadban virus was purified from the plaques by three passage cycles, we cannot exclude contamination at this point with some Gram-positive-specific phage/s that could contribute to the lysis of the intact Arthrobacter sp. cells."